# Numerical Simulation of the Effect of Solidified Shell Conductivity and Billet Sizes on the Magnetic Field with Final Electromagnetic Stirring in Continuous Casting

**DOI:** 10.3390/ma16134765

**Published:** 2023-07-01

**Authors:** Guofang Xu, Ruisong Tan, Bo Song, Wei Liu, Shufeng Yang, Xiaotan Zuo, Yan Huang

**Affiliations:** 1Metallurgical and Ecological Engineering School, University of Science and Technology Beijing, Beijing 100083, China; xgf2672@163.com (G.X.); d202210109@xs.ustb.edu.cn (R.T.); yangshufeng@ustb.edu.cn (S.Y.); 2Steelmaking Department, Wuhu Xinxing Ductile Pipe Co., Ltd., Wuhu 241002, China; zuoxiaotan345@163.com (X.Z.); huangyanyejin@163.com (Y.H.)

**Keywords:** F-EMS, numerical simulation, round billet sizes, electromagnetic field, solidified shell conductivity

## Abstract

Coupled with the results of a 2D heat transfer model, a 3D electromagnetic stirring round billet model is developed, which is considered for the difference in the conductivity of solidified shell and molten steel. The electromagnetic field distribution features of the billet and the effect of round billet sizes on the electromagnetic field are investigated. It is found that as the solidified shell conductivity of the Φ600 mm round billet increases from 7.14 × 10^5^ S·m^−1^ to 1.0 × 10^6^ S·m^−1^, the magnetic induction intensity decreases and the maximum value of electromagnetic force drops from 7976.26 N·m^−3^ to 5745.32 N·m^−3^. The magnetic induction intensity on the center axis of the stirrer rarely changes in the range of Φ100–Φ200 mm. With the increase in the round billet from Φ300 mm to Φ600 mm, the magnetic induction intensity and the electromagnetic force on the center axis of the stirrer decrease slowly and then significantly. In the range of 2–8 Hz, as the current strength reaches its maximum, the electromagnetic force can be increased by increasing the current frequency for round billets of Φ100–Φ500 mm, while there is an optimal current frequency for round billets larger than Φ600 mm.

## 1. Introduction

Recently, large billets have been extensively applied in large-scale equipment, including wind power generation, large or extremely large-scale ring products, the petrochemical industry, and other equipment [1]. Nevertheless, the large round billet in continuous casting is prone to problems such as center porosity [2,3], shrinkage cavity [4], and macrosegregation [1,5] at the end of the solidification. The mold electromagnetic stirring (M-EMS) and the final electromagnetic stirring (F-EMS) [6,7,8] are popularly applied in the production of continuous casting of round billets to solve the above problems.

A large number of efforts have been devoted to research on the F-EMS installation position, magnetic field, fluid flow, heat, and solute transfer in continuous casting by numerical simulation. Ayata et al. [9] and Mizukami et al. [10] investigated the central solid phase rate of the F-EMS at the optimal installation position. Liu et al. [11] and Li et al. [12] focused on fluid flow and heat transfer of the continuous casting strand with F-EMS. Sun et al. [5], Fang et al. [13], and Dong et al. [14] studied the effect of F-EMS on fluid flow, heat transfer, and mass transfer. Li et al. [15] developed a three-dimensional model based on segregation, considering the F-EMS installation position, magnetic field, fluid flow, heat, and solute transfer at the final stage of solidification. However, the above investigations rarely discussed the electromagnetic field of the large round billet. Although Ren et al. [16] studied the distribution of electromagnetic force and the effect of current intensity and frequency, as well as Wan et al. [17], who investigated the effect of electromagnetic force on carbon distribution, it is necessary to further consider the difference in conductivity between the solidified shell and the molten steel and the influence of the billet sizes on the electromagnetic field. Hence, in the current work, coupled with 2D heat transfer model results, a 3D electromagnetic stirring round billet model is developed to simulate the electromagnetic field distribution features of the billet, which are considered for the difference in the conductivity of solidified shell and molten steel and the effect of different dimensions on the electromagnetic field, providing the theoretical guidance for F-EMS.

## 2. Mathematical Model

### 2.1. Basic Assumption

Due to the complex, high-temperature, and physical processes of the continuous casting process, the model of F-EMS and the heat transfer partial differential solidification mathematical model are simplified by the following assumptions [18,19,20]:(1)Ignoring the displacement current, the electromagnetic field is considered as the magnetic quasi-static field in the low-frequency stirring condition;(2)During electromagnetic stirring, molten steel is regarded as stationary, and the influence of the steel movement on the electromagnetic field is neglected;(3)The steel is assumed to be an incompressible conductive fluid;(4)In the model of F-EMS, the steel of the liquid core is seen as a circular platform body, while the length of the steel of the liquid core and the thickness of the billet solidified shell are calculated by ProCAST 2018 software;(5)Since the heat transfer in the cross-section direction of the continuous casting round billet is much greater than the heat transfer along the drawing direction, the heat transfer along the drawing direction of the casting round billet can be neglected;(6)In the continuous casting second cooling zone, the surface of the round billet cooled uniformly;(7)The effect of the steel flow on the internal heat transfer is not considered;(8)Ignoring the influence of the crystallizer vibration and other factors, the casting temperature is equal to the tundish temperature of the crystallizer;(9)The contact heat transfer and the surface radiation between the round billet and the support rolls are used for the integrated thermal conductivity.

### 2.2. Control Equations

#### 2.2.1. Electromagnetic Field

The electromagnetic field distribution is described by Maxwell’s system of equations, which is simplified in this study by neglecting the effect of displacement currents with the following equations [18,19].
(1)∇×H=J
(2)$$\nabla \times E=-\frac{\partial B}{\partial t}$$
(3)∇×B=0
(4)B=μH
(5)J=σE
where H is the magnetic field intensity in A·m^−1^; J is the current density in A·m^−2^; E is the electric field intensity in V·m^−1^; B is the magnetic induction intensity in T; t is the time in seconds; μ is Magnetic permeability in H·m^−1^; σ is the electric conductivity in S·m^−1^.

The electromagnetic force uses the time-averaged Lorentz force in solving and analysis calculated by the following equation:(6)Fmag=12ReJ×B
where $F_{\mathrm{mag}}$ is the average value of electromagnetic force time, N·m^−3^; $B^{*}$ is the conjugate complex number of magnetic flux density *B*.

#### 2.2.2. Transfer Behavior

The mathematical model of the heat transfer equation is as follows [20]:(7)$$\frac{\partial T}{\partial t}=\frac{\lambda }{\rho C_{p}}\left(\frac{\partial^{2}T}{\partial x^{2}}+\frac{\partial^{2}T}{\partial y^{2}}\right)$$
where *T* is the local temperature in K; ρ is the density in kg·m^−3^; λ is the thermal conductivity in W·m^−1^·K^−1^; $C_{p}$ is specific heat in J·kg^−1^·K^−1^.

### 2.3. Boundary Conditions

#### 2.3.1. Electromagnetic Field

(1)The F-EMS uses three pairs of coil windings loaded with three-phase alternating current, with a phase difference of 120° for each phase.(2)The magnetic lines of force are parallel to the surface of the air unit enclosed outside the stirring.(3)The boundary condition between the coil and the core of the electromagnetic stirring is established as the insulation.

#### 2.3.2. Flow and Solidification

(1)The inlet temperature at the mold is equal to the casting temperature of the molten steel.(2)As the heat transfer coefficient of the 2D model varies with cooling zones, according to the above assumptions, the secondary cooling boundary conditions of each section are set as follows [21].
(a)The mold
(8)$$q_{m}=\frac{\rho C_{p}Q\Delta T}{S}$$
where q_m_ is the average heat flux at the surface of the casting billet in W·m^−2^; ρ is the density of mold cooling water in kg·m^−3^; *Q* is the cooling water flow rate in m^3^·s^−1^; ΔT is the temperature difference between the inlet and outlet of the mold cooper in K; *S* is the effective contact area between the mold cooper and the casting billet in m^2^.(b)The secondary cooling zone
(9)$$h_{1}=581\omega^{0.541}\left(1-0.0045T_{w}\right)$$
(10)$$h_{2}=350\omega +150$$
(11)$$\omega =\frac{R\upsilon_{c}S\gamma \xi_{n}}{60\pi DL_{n}}$$
where h_1_ and h_2_ are the heat transfer coefficient of the secondary cooling zone in W·m^−2^·K^−1^; ω is the water density in L·m^−2^·S^−1^; *T*_w_ is the temperature of the spray cooling flux in K; R is the water ratio of the continuous cooling water in L·kg^−1^; $\upsilon_{c}$ is the casting speed in m·min^−1^; *S* is the cross-sectional area of the billet in m^2^; γ is the density of the steel in kg·m^−3^; $\xi_{n}$ is the water distribution ratio of the second cooling zone in %; $L_{n}$ is the length of the nth zone in m.(c)The air-cooling zone
(12)$$q=\varepsilon \sigma \left(T^{4}-T_{w}^{4}\right)$$
where σ is the Stefan–Boltzmann constant, 5.67 × 10^−8^ W·(m^2^·K^4^)^−1^; ε is steel emissivity; *T*_w_ is the ambient temperature in K.

### 2.4. Solution Method

In this paper, a 2D–3D model was developed and divided into two parts for calculation. Initially, the solidified shell thickness of solidification and the length of the liquid core were available through the solidification heat transfer model of ProCAST 2018 software. A tetrahedral mesh with a grid number of 53,415 was used by the moving slice model. After that, the electromagnetic field of the F-EMS was simulated numerically by using the finite software Ansys Electronics 2021. A tetrahedral mesh with a grid number of 446,629 was applied by the model of F-EMS.

## 3. Geometric Model

The section size of the continuous casting billet is Φ600 mm, and the radius of the continuous casting arc section is 17 m. Moreover, the total height and effective height of the mold are 0.80 m and 0.69 m, and the lengths of the secondary cooling zone are 0.24 m, 0.811 m, and 1.22 m, respectively. The moving slice model is developed by Ansys SpaceClaim 2021 software, with a model thickness and diameter of 10 mm and 600 mm, respectively, as shown in the moving slice Figure 1.

According to the continuous casting conditions, as follows: casting speed is 0.26 m·min^−1^, superheat is 30 °C, and specific water flow is 0.12 L·kg^−1^, respectively, the solidified shell thickness of solidification and the length of the liquid core are measured as 170.7 mm and 258.6 mm, respectively, at 11.7 m the solidification contour, as shown in Figure 2a, and are measured as 194.5 mm and 211 mm, respectively, at 13.7 m the solidification contour, which is shown in Figure 2b.

Based on the actual F-EMS equipment of a mill, the solidified shell thickness of solidification, and the length of the liquid core of the moving slice model, the F-EMS model is established with the finite software of Ansys Electronics, as seen in Figure 3. The electromagnetic stirring consists mainly of a coil, a core, a billet solidified shell, a liquid core, and a closed airfield not displayed, and the stirring is a salient pole structure with six coils and three pairs of windings. The stirring has an inner diameter of 850 mm, an outer diameter of 1280 mm, and a height of 880 mm. The center of the stirring is 12.7 m away from the meniscus, and the type of stirring is continuous. The casting billet is built by Ansys Electronics in the section sizes Φ100 mm, Φ200 mm, Φ300 mm, Φ400 mm, Φ500 mm, and Φ600 mm, and then the electromagnetic fields are simulated. Moreover, Table 1 shows the material physical properties and, mainly, continuous casting process parameters of F-EMS.

## 4. Model Validation

In this study, to verify the accuracy of the heat transfer model, the continuous casting conditions are as follows: casting speed 0.26 m·min^−1^, superheat 30 °C, and specific water flow 0.12 L·kg^−1^, respectively. In on-site production, the surface temperature of the casting billet is measured by an infrared temperature gun and compared with the simulated temperature. As can be seen from Figure 4a, there is a small difference between the simulated temperature field and actual temperature results, which verifies the accuracy of the model temperature field.

Using other numerical solutions as reference values to verify one’s model is one of the model validation methods. Hence, the results of the magnetic induction intensity at the center of the stirrer under the different current intensities of F-EMS studied by Ren et al. [16] in this work are taken to verify the accuracy and validity of the present model. As can be seen from Figure 4b, the difference between this model and data from Ren et al. on magnetic induction intensity at the center of the stirrer under different current intensities is not significant, with a maximum difference in magnetic induction intensity of only 4.25 mT, which verifies the accuracy and validity of the present model.

## 5. Results and Discussion

### 5.1. Effect of Current Frequency and Intensity on Electromagnetic Field for the Round Billet

In this section, the bulk conductivities of the solidified shell and liquid core are 1.0 × 10^6^ and 7.14 × 10^5^, respectively. The distribution pattern of the simulated values for the magnetic field in the center axis of the stirrer with and without the strand at a current intensity of 350 A and a frequency of 4 Hz is shown in Figure 5. As can be seen from Figure 5, the magnetic induction intensity shows a symmetrical distribution, which increases and then decreases in the direction of the centerline of the F-EMS stirrer, reaching its maximum value at the center of the stirrer. The simulation value of the magnetic induction intensity without the strand is slightly larger than the simulation value of the magnetic induction intensity of the strand, with the difference in maximum magnetic induction intensity being only 2.07 mT, which indicates that the strand’s conductivity has a certain shielding influence on the magnetic induction intensity.

Figure 6 shows the contours of the magnetic induction intensity at the surface of the liquid core of the round billet Figure 6a and the electromagnetic force density on the transverse section Figure 6b at a current intensity of 500 A and a frequency of 4 Hz. It has been observed that the magnetic induction intensity on the surface of the liquid core shows a distribution with a large value in the middle and a small value at both edges, with the maximum magnetic induction intensity in the center of the stirrer decreasing rapidly after the stirrer. As can be seen from Figure 6b, the electromagnetic force is distributed circumferentially in the cross-section at the liquid core in the center of the F-EMS and increases with the distance from the center of the cross-section of the strand. The majority of the electromagnetic force is predominantly applied to the solidified shell, with the less effective electromagnetic force working on the liquid core owing to the lower liquid pool in the late solidification.

Figure 7a shows the distribution of the magnetic induction intensity on the central axis at different current frequencies with a current intensity of 500 A. It can be seen that the magnetic induction intensity on the central axis decreases with increasing frequencies. The decline in magnetic induction intensity is associated with the skin depth and the shielding effect of the solidified shell with a certain conductivity on the magnetic field. The skin depth can be given by Equation (13) [12].
(13)$$\delta =\sqrt{\frac{1}{\pi f\mu \sigma }}$$
where δ is skin depth in m; μ is magnetic permeability in H·m^−1^; *f* is the current frequency in Hz; σ is the electric conductivity in S·m^−1^.

When the current frequency is 2–8 Hz and the conductivity of the solidified shell is 1 × 10^6^ S·m^−1^, the skin depths are calculated by Equation (13) as 355.88, 251.65, 205.47, and 177.94 mm, respectively. The increase in current frequency decreases the skin depth, leading to a reduction of the magnetic field penetration depth. In addition, the solidified shell with a certain conductivity has a shielding effect; the greater the frequency, the stronger the shielding [12,23]. These two reasons led to the magnetic induction intensity at the center of the stirrer decreasing from 80.8 mT to 54.30 mT while the current frequency increased from 2 Hz to 8 Hz.

Figure 7b shows the distribution of the magnetic induction intensity on the central axis at different current intensities with a current frequency of 4 Hz. It can be observed that the magnetic induction intensity increases with the increase in current intensity at the same position as the stirring center, and the magnetic induction intensity is maximum near the center of the stirrer. If the current intensity is 100, 200, 300, 400, and 500 A, the magnetic induction intensity at the center of the stirrer is 14.40, 28.93, 43.39, 58.04, and 72.50 mT, respectively. For each 100 A increase in current intensity, the magnetic induction intensity at the center of the stirrer increases by approximately 14.52 mT.

The electromagnetic force with magnetic induction intensity and frequency is defined by the empirical Equation (14) [24].
(14)$$F\propto B^{2}f$$
where *B* is the magnetic induction intensity in the strand in T; *f* is the current frequency in Hz. The electromagnetic force is related to the magnetic induction intensity as a quadratic function and linear in frequency.

The contours of electromagnetic force distribution on the liquid core transverse section with a current of 500 A and a frequency from 2 Hz to 8 Hz are presented in Figure 8. It can be seen that the electromagnetic force first increases and then slowly decreases as the frequency increases. The electromagnetic force increases with distance from the center of the strand, while it is smaller at the center of the liquid core. From Equation (14), it can be found that when the current frequency is 8 Hz, the electromagnetic force raised by the increase in current frequency is lower than the weakening effect on the electromagnetic force caused by the decrease in magnetic induction intensity.

The contours of electromagnetic force distribution on the liquid core transverse section with a frequency of 4 Hz and current intensity ranging from 100 A to 500 A are presented in Figure 9. As can be observed from Equation (14), the electromagnetic force increases significantly with the increase in current intensity due to the quadratic function relationship between electromagnetic force and current intensity. The maximum electromagnetic force in the liquid core zone is 200.97, 760.00, 1710.00, 3221.73, and 5025.30 N·m^−3^ at the current intensities of 100, 200, 300, 400, and 500 A, respectively.

### 5.2. Effect of Solidified Shell Conductivity on Electromagnetic Field for the Round Billets

In this section, the bulk conductivity of the solidified shell and liquid core is defined as Case1 and Case2. In Case1, the bulk conductivities of solidified shell and liquid core are 1.0 × 10^6^ S·m^−1^ and 7.14 × 10^5^ S·m^−1^, respectively, and in Case2, the bulk conductivities of solidified shell and liquid core are both set to 7.14 × 10^5^ S·m^−1^.

The distribution of magnetic induction intensity at the center of the stirrer with different solidified shell conductivities is presented in Figure 10. It can be seen that the central magnetic induction intensity for solidified shell conductivity at 1 × 10^6^ S·m^−1^ is less than the central solidified shell conductivity at 7.14 × 10^5^ S·m^−1^ at the same current frequency. As the current frequency increases, the difference in magnetic induction intensity at the center of the stirrer increases for different shell conductivities.

Figure 11 shows the contours of electromagnetic force distribution on the liquid core transverse section under different current frequencies for the solidified shell at 7.14 × 10^5^ S·m^−1^. It can be observed from Figure 11 that as the current frequency increases, the electromagnetic force on the cross-section of the liquid core increases significantly and then increases slowly, reaching a maximum value of 7976.26 N·m^−3^ at a current frequency of 8 Hz, whereas, as shown in Figure 8, the electromagnetic force on the cross-section of the liquid core increases and then decreases slightly with the increase in frequency at the solidified shell conductivity of 1 × 10^6^ S·m^−1^, achieving a maximum value of 5745.32 N·m^−3^. So the simulation results show that when the current intensity reaches its maximum value at 500 A if further increase in electromagnetic force is desired, the optimal frequency can be obtained for the round billet of Φ600 mm at a current frequency of 6 Hz.

### 5.3. Effect of Round Billets Size on Electromagnetic Field

In this section, the tendency of magnetic induction intensity and electromagnetic force of round billets in the range of Φ100–Φ600 mm with a current frequency of 2–8 Hz and a current intensity of 500A is studied to provide theoretical guidance for round billet production. As the corresponding liquid core length and solidified shell thickness can’t be measured for Φ100–Φ500 mm round billets, the casting billet conductivity is set to the liquid steel conductivity for all investigations on the effect of the round billet size on the electromagnetic field. The bulk conductivity of solidified shell and liquid core of Φ600 mm round billets are 1.0 × 10^6^ and 7.14 × 10^5^, respectively. The distribution of magnetic induction intensity on the central axis of the stirrer and the center of the stirrer for different round billet sizes at a current frequency of 4 Hz is presented in Figure 12. As can be seen from Figure 12a, the magnetic induction intensity on the center axis of the stirrer decreases slightly with the round billet size from Φ100 mm to Φ400 mm, whereas the magnetic induction intensity on the center axis of the stirrer decreases significantly if the round billet size is larger than Φ400 mm. As shown in Figure 12b, the magnetic induction intensity at the center of the round billet stirrer decreases as the diameter of the round billet keeps increasing. When the size of the round billet increases from Φ100 mm to Φ400 mm, the variation in magnetic induction intensity at the center of the stirrer is less than 1 mT. As the round billet increases from Φ400 mm to Φ500 mm, the magnetic induction intensity at the center of the stirrer decreases by 2.33 mT. Moreover, With the increase in billet from Φ500 mm to Φ600 mm, the magnetic induction intensity decreases by 3.67 mT.

Figure 13 shows the distribution of magnetic induction intensity on the central axis of the stirrer at different round billet sizes and frequencies at 500A. When the current frequency is 2–8 Hz and the conductivity of the solidified shell and liquid core is 7.14 × 10^5^ S·m^−1^, the skin depths are calculated by Equation (13) as 421.16, 297.81, 243.16, and 210.58 mm, respectively. As observed in Figure 13, for large round billets Φ100 mm and Φ200 mm, the magnetic induction intensity on the center axis of the stirrer hardly differs with increasing current frequency since the minimum value of the skin depth of the round billet is 210.58 mm at the current frequency of 2–8 Hz, which enables the magnetic field to penetrate the whole round billets. If the billet is in the range of Φ300–Φ600 mm, the magnetic induction intensity on the center axis of the stirrer decreases as the current frequency increases. With the increase in billet diameter, the magnetic induction intensity on the center axis of the stirrer decreases remarkably. The maximum decrease in magnetic induction intensities of Φ300–Φ600 mm round billets are 0.62 mT, 1.67 mT, 4.12 mT, and 8.86 mT. In the range of 2–8 Hz, the magnetic induction intensities of Φ100–Φ500 mm round billets decrease less than the magnetic induction intensity of 600 mm, considering the different conductivities of solidified shell and liquid core. Hence, it can be calculated by the empirical Equation (14) that the electromagnetic force increases with the increase in current frequency.

Since the electromagnetic force is proportional to the value of B^2^*f*, Figure 14 shows the value of B^2^*f* calculated under the empirical Equation (14) for different billet sizes and frequencies. It can be seen that the electromagnetic force decreases with the increase in billet size at the same current frequency; the larger the round billet size, the more obvious the decrease. When the current frequency is 2–8 Hz, the electromagnetic force increases with the increase in current frequency in the range of Φ100–Φ500 mm, while the electromagnetic force increases and then decreases with the increase in current frequency for the round billet of Φ600 mm, which is consistent with the simulation results in Section 5.1. For round billets Φ100–Φ500 mm, if the stirring capacity of F-EMS is to be improved, firstly, the current intensity should be increased, and then the frequency should be increased [12]. However, for round billets larger than Φ600 mm, due to the increased thickness of the solidified billet shell and current frequency, the skin depth decreases and the solidified billet shell shielding is enhanced, resulting in a decreased stirring capacity of F-EMS. If the stirring capacity of F-EMS is also to be enhanced, the current intensity must first be increased to obtain the optimal current frequency [23].

## 6. Conclusions

(1)With the increase in the frequency from 2 Hz to 8 Hz, the magnetic induction intensity on the central axis decreases, while the electromagnetic force on the transverse section of the liquid core initially increases and then decreases, reaching a maximum value of 5745.32 N·m^−3^ on the cross-section of the liquid core at the current frequency of 6 Hz. As the current frequency increases from 100 A to 500 A, the magnetic induction intensity and the electromagnetic force on the transverse of the liquid core increase.(2)With the rise of the solidified shell conductivity from 7.14 × 10^5^ S·m^−1^ to 1.0 × 10^6^ S·m^−1^, the magnetic induction intensity and electromagnetic force at the liquid core decrease at the same current frequency. As the current frequency increases, the difference between the stirrer center magnetic induction intensity and electromagnetic force increases for different solidified shell conductivities. The results of the simulation show that the optimal current frequency and current intensity of F-EMS are 6 Hz and 500 A for a Φ600 mm round bloom.(3)The simulation results only consider the conditions of current frequency 2–8 Hz and round billets Φ100–Φ600 mm. In the range of round billets Φ100–Φ500 mm, the electromagnetic force increases with the rise of current frequency, while for billet Φ600 mm, the electromagnetic force increases and then decreases with the increase in current frequency. The magnetic induction intensity on the center axis of the stirrer rarely changes in the range of Φ100–Φ200 mm of the billets. When the current frequency is 2–8 Hz, the magnetic induction intensity on the center axis of the stirrer decreases slowly and then significantly as the round billet increases from Φ300 mm to Φ600 mm.

## Figures and Tables

**Figure 1 materials-16-04765-f001:**
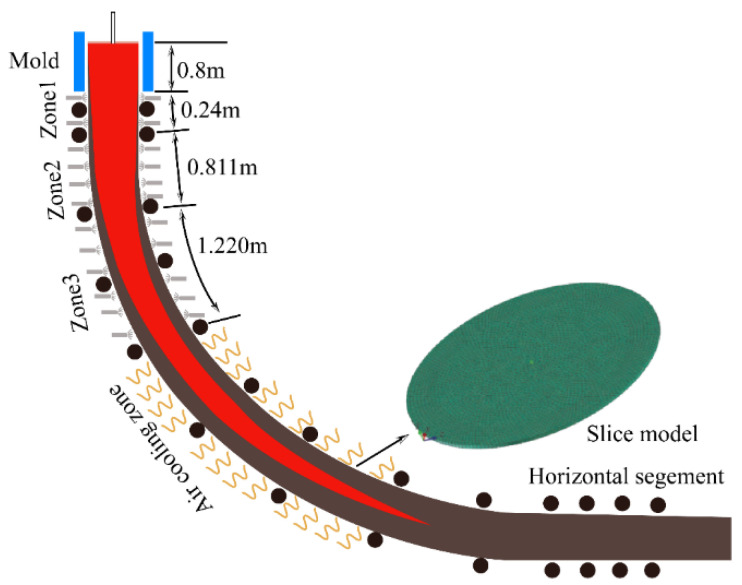
Moving slice model.

**Figure 2 materials-16-04765-f002:**
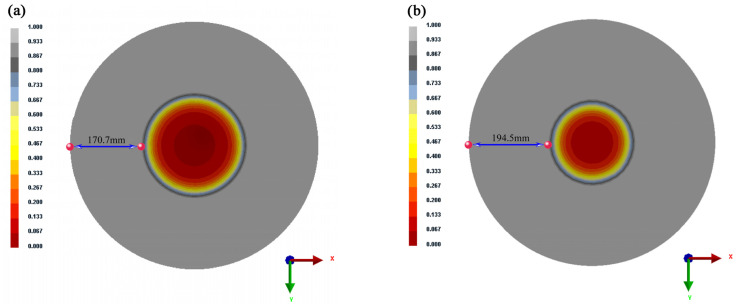
Solidification contour plot of moving slices (**a**) at 11.7 m and (**b**) at 13.7 m.

**Figure 3 materials-16-04765-f003:**
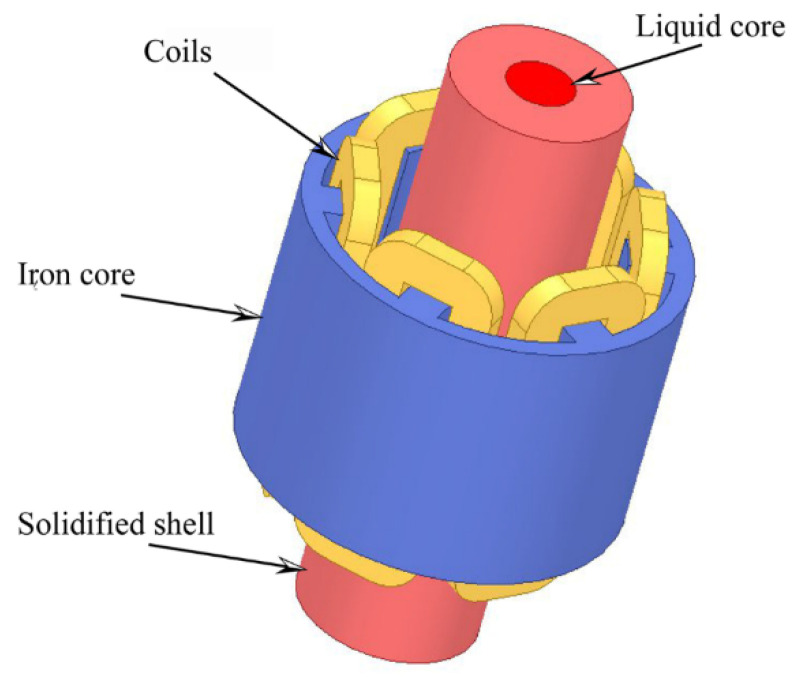
Model of the F-EMS.

**Figure 4 materials-16-04765-f004:**
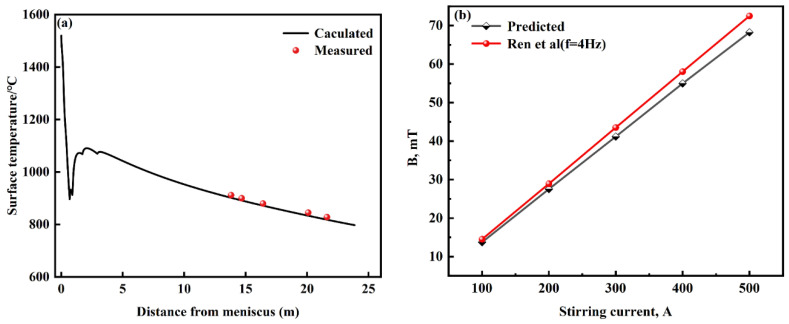
Comparison between simulated and measured values of the surface temperature of the round billet (**a**) and the magnetic induction intensity along the central axis of the stirrer (**b**).

**Figure 5 materials-16-04765-f005:**
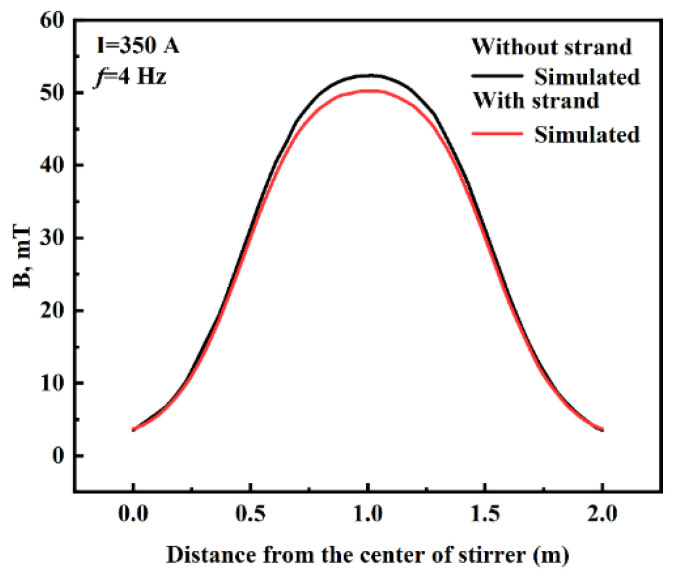
The distribution of magnetic induction intensity along the central axis of the stirrer.

**Figure 6 materials-16-04765-f006:**
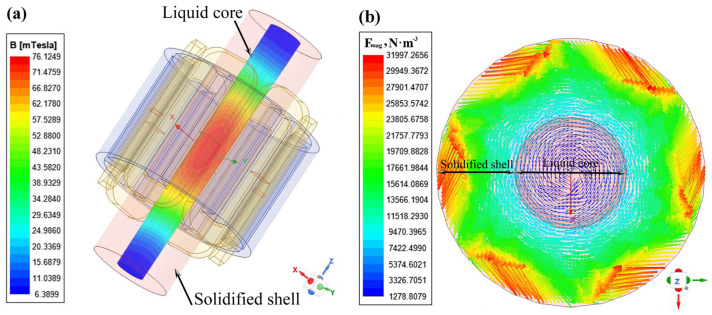
Contours of the magnetic induction intensity at the surface of the liquid core of the round billet (**a**) and electromagnetic force density on the transverse section (**b**).

**Figure 7 materials-16-04765-f007:**
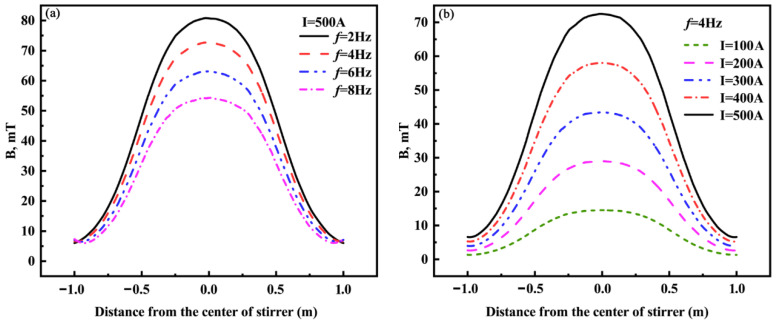
Distribution of magnetic induction intensity on the center axis of the stirrer at different current frequencies (**a**) and current intensities (**b**).

**Figure 8 materials-16-04765-f008:**
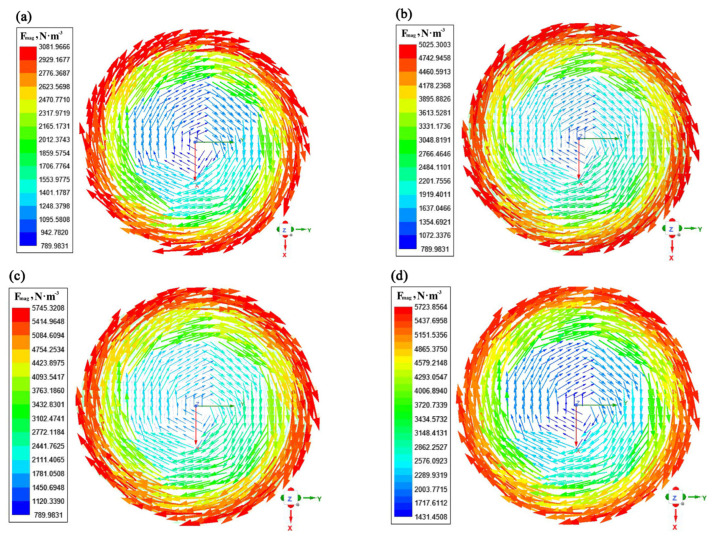
Contours of electromagnetic force distribution on the liquid core transverse section under different current frequencies: (**a**) I = 500 A, *f* = 2 Hz; (**b**) I = 500 A, *f* = 4 Hz; (**c**) I = 500 A, *f* = 6 Hz; (**d**) I = 500 A, *f* = 8 Hz.

**Figure 9 materials-16-04765-f009:**
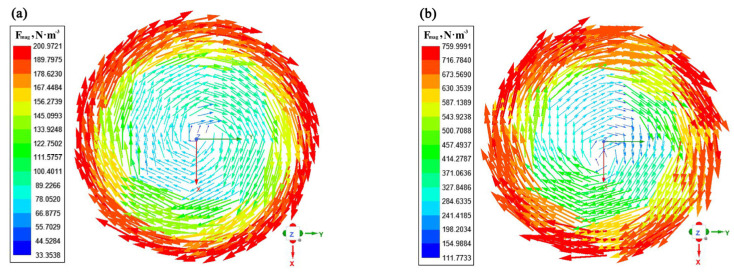
Contours of electromagnetic force distribution on the liquid core transverse section under different current intensities: (**a**) I = 100 A, *f* = 4 Hz; (**b**) I = 200 A, *f* = 4 Hz; (**c**) I = 300 A, *f* = 4 Hz; (**d**) I = 400 A, *f* = 4 Hz; (**e**) I = 500 A, *f* = 4 Hz.

**Figure 10 materials-16-04765-f010:**
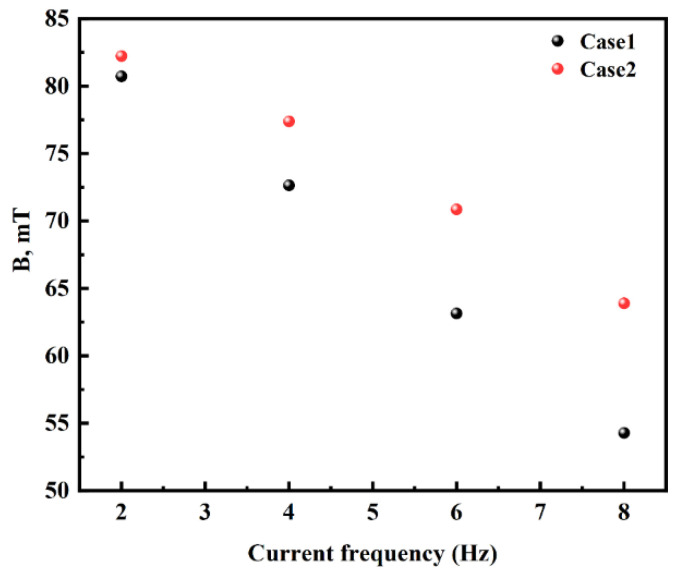
Distribution of magnetic induction intensity at the center of the stirrer with different solidified shell conductivities.

**Figure 11 materials-16-04765-f011:**
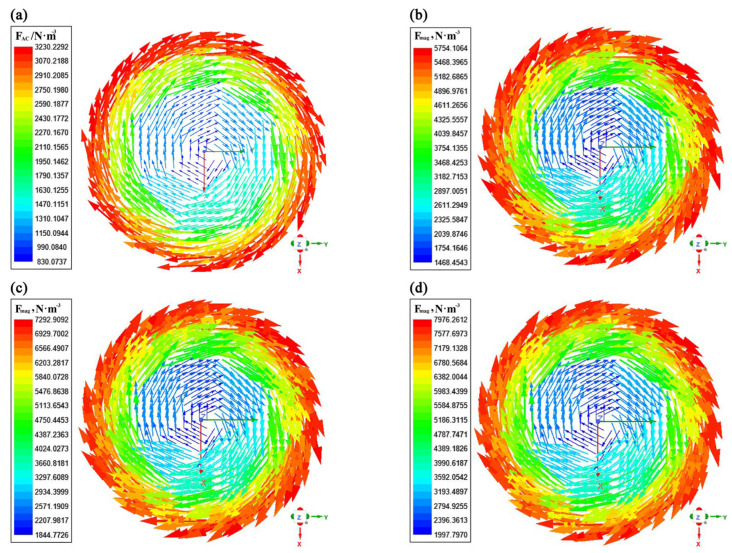
Contours of electromagnetic force distribution on the liquid core transverse section under different current frequencies: (**a**) I = 500 A, *f* = 2 Hz; (**b**) I = 500 A, *f* = 4 Hz; (**c**) I = 500 A, *f* = 6 Hz; (**d**) I = 500 A, *f* = 8 Hz.

**Figure 12 materials-16-04765-f012:**
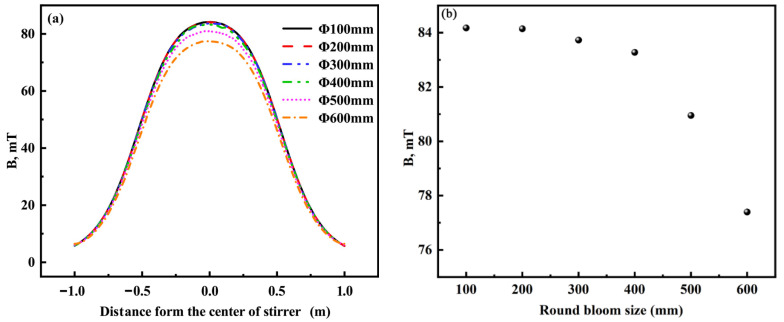
Distribution of magnetic induction intensity on the center axis of the stirrer (**a**) and the center of the stirrer (**b**) for different round billet sizes.

**Figure 13 materials-16-04765-f013:**
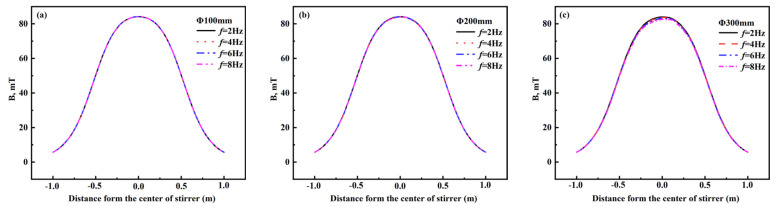
Distribution of magnetic induction intensity on the stirrer central axis of the stirrer at different frequencies for round billets of Φ100–Φ600 mm at 500 A. (**a**) Φ100 mm; (**b**) Φ200 mm; (**c**) Φ300 mm; (**d**) Φ400 m; (**e**) Φ500 mm; (***f***) Φ600 mm.

**Figure 14 materials-16-04765-f014:**
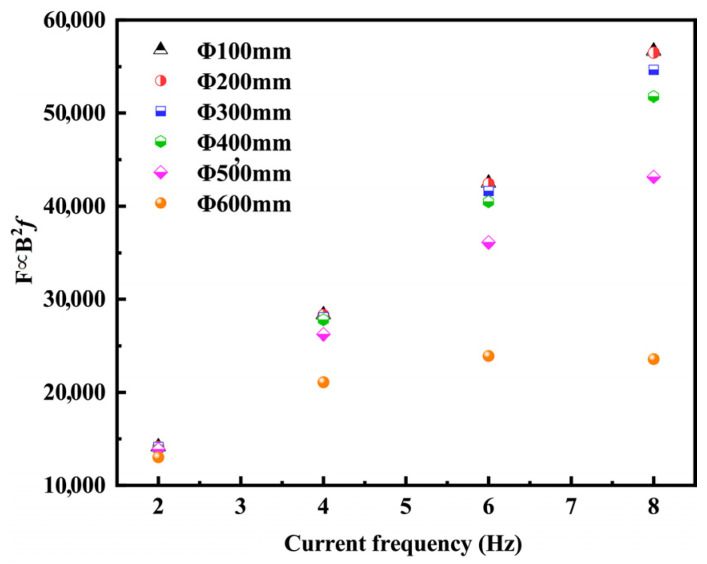
Calculated values for different round billet sizes and frequencies with the empirical equation for magnetic force.

**Table 1 materials-16-04765-t001:** Material physical properties and main process parameters of the continuous casting process.

Parameters	Value
Round billet Diameter (mm)	600
Casting Speed (m·min^−1^)	0.26
Superheat (°C)	30
Current Frequency (Hz)	2–8
Current intensity (A)	100–500
Density of liquid steel (kg·m^−3^)	7020
Relative magnetic permeability of liquid core, solidified shell, air, and coil	1.0
Relative magnetic permeability of iron core	1000
Bulk conductivity of liquid core (S·m^−1^)	7.14 × 10^5^
Bulk conductivity of solidified shell [22] (S·m^−1^)	1.0 × 10^6^
Bulk conductivity of air (S·m^−1^)	8.855 × 10^−6^

## Data Availability

The data presented in this study are available on request from the corresponding author. The data are not publicly available due to involving trade secrets.

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
