# Peer review of "Numerical Simulation of the Effect of Solidified Shell Conductivity and Billet Sizes on the Magnetic Field with Final Electromagnetic Stirring in Continuous Casting"

_materials, 2023, doi:10.3390/ma16134765_

Round 1

Reviewer 1 Report

The authors of the manuscript simulate the electromagnetic stirring in continuous casting. The model is validated prior to the investigation of important parameters, including current, frequency and geometry. The results are useful for technical point of view. Only minor revision is required.

1) The citations are mostly in the introduction but lacking in the discussion. The manuscript will improve by comparing the present results to the literature. The end of discussion should cover the implementation or suggestion.

2) Limitations of the simulation should be added in the conclusion

3) In the first paragraph of 5.2, the unit of conductivity is missing.

4) Some errors in figures, e.g. Disrance Fig 12(a), Fig9(f) (should be (e)). Figures 13(a)-(f) are not properly clarified in the caption. All figures’ resolution is not good but will presumably be improved on the published version.  

5) There are other inconsistencies, particularly in the reference list. For example, the use of capital letters in article and journal names. Country of journal’s origin should be left out.

The writing is comprehensible, but the manuscript still needs language editing. Take the final paragraph of the discussion section as an example.

"Whether it is small round billets or large round billets if the stirring capacity of the electromagnetic force is required to be increased, it can be done two things. For one thing: it can be done by enlarging the current intensity. For the other thing: when the current strength reaches the upper limit, a certain electromagnetic stirring intensity can be improved by using the higher current frequency for small round billets[12], whereas the appropriate current frequency should be selected to increase the strengths of electromagnetic stirring for large round billet."

Author Response

Dear reviewers:

Thank you very much for your review and comments on our manuscript entitled “Numerical Simulation of Effect of Solidified Shell Conductivity and Billet Sizes on Magnetic Field with Final Electromagnetic Stirring in Continuous Casting” (Manuscript Number:materials-2390926). These comments are very helpful for improving the manuscript, as well as of great guiding significance for our further research. We have revised the manuscript according to your kind advices. The revisions are marked in red in the revised manuscript.

Please find the following responses to the comments:

Reviewer1#

The authors of the manuscript simulate the electromagnetic stirring in continuous casting. The model is validated prior to the investigation of important parameters, including current, frequency and geometry. The results are useful for technical point of view. Only minor revision is required.

1) The citations are mostly in the introduction but lacking in the discussion. The manuscript will improve by comparing the present results to the literature. The end of discussion should cover the implementation or suggestion.

Response:Thanks very much for the comment. The suggestions made have been revised and marked in red in the original text.

2) Limitations of the simulation should be added in the conclusion

Response:Thanks very much for the comment. The limitations of the simulation had been modified and were marked in red.

3) In the first paragraph of 5.2, the unit of conductivity is missing.

Response:Thanks very much for the comment. The errors had been modified and were marked in red.

4) Some errors in figures, e.g. Disrance Fig 12(a), Fig9(f) (should be (e)). Figures 13(a)-(f) are not properly clarified in the caption. All figures’ resolution is not good but will presumably be improved on the published version.

Response:Thanks very much for the comment. The errors had been modified and were marked in red.

5) There are other inconsistencies, particularly in the reference list. For example, the use of capital letters in article and journal names. Country of journal’s origin should be left out.

Response:Thanks very much for the comment. The errors of the reference had been modified and were marked in red.

Comments on the Quality of English Language

The writing is comprehensible, but the manuscript still needs language editing. Take the final paragraph of the discussion section as an example.

"Whether it is small round billets or large round billets if the stirring capacity of the electromagnetic force is required to be increased, it can be done two things. For one thing: it can be done by enlarging the current intensity. For the other thing: when the current strength reaches the upper limit, a certain electromagnetic stirring intensity can be improved by using the higher current frequency for small round billets[12], whereas the appropriate current frequency should be selected to increase the strengths of electromagnetic stirring for large round billet."

Response Thanks very much for the comment. the Quality of English Language had been modified and were marked in red.

Reviewer 2 Report

Review of the manuscript "Numerical Simulation of Effect of Solidified Shell Conductivity and Billet Sizes on Magnetic Field with Final Electromagnetic Stirring in Continuous Casting".

In this work, a three-dimensional model of a round billet with electromagnetic stirring has been developed, which takes into account the difference in the conductivity of the solidified crust and molten steel. The features of the distribution of the electromagnetic field of the workpiece and the influence of the dimensions of the round workpiece on the electromagnetic field are studied.

There are a few notes about the work.

1. The symbol F is used to denote force in equation (6). Equation (8) uses the same symbol to denote area.

2. Fig. 4a. In the text of the work, the authors talk about the good convergence of the calculated and experimental data. However, for the range from 0 to 25 m, there are only four experimental points in the range of 13-22 m.

3. It is not clear the temperature of which point of the ingot section was calculated - surface / center? What's with the dip at the beginning of the graph?

4. In the first paragraph of section 5.1. and in several other places in the text, conductivity values are given without units of measure.

5. As the main formula for calculating the electromagnetic force, the authors, with reference to [24], took the expression F≈B2f. This is not entirely clear. In [24] itself, the equation

T=0.25πσω B2R4L

But despite the fact that the main goal of the work under review indicated by authors as necessary to establish the effect of the difference in conductivity between the hardened crust and molten steel and the effect of the dimensions of the workpieces on the electromagnetic field, these parameters were not taken into account in the calculation of the force. Why?

6. It can be seen from the text of the work that the calculations were carried out for a specific production line. However, there are no practical recommendations for production in the conclusion.

Author Response

Dear reviewers:

Thank you very much for your review and comments on our manuscript entitled “Numerical Simulation of Effect of Solidified Shell Conductivity and Billet Sizes on Magnetic Field with Final Electromagnetic Stirring in Continuous Casting” (Manuscript Number:materials-2390926). These comments are very helpful for improving the manuscript, as well as of great guiding significance for our further research. We have revised the manuscript according to your kind advices. The revisions are marked in red in the revised manuscript.

Please find the following responses to the comments:

Reviewer2#

There are a few notes about the work.

  1. The symbol F is used to denote force in equation (6). Equation (8) uses the same symbol to denote area.

Response:Thanks very much for the comment. F in equation (8) is modified to S.

  1. Fig. 4a. In the text of the work, the authors talk about the good convergence of the calculated and experimental data. However, for the range from 0 to 25 m, there are only four experimental points in the range of 13-22 m.

Response:Thanks very much for the comment. This is caused by the equipment on site. At 0-12.7m, infrared temperature gun is difficult to measure because the whole equipment of continuous casting is closed, after 12.7 m, the equipment is not enclosed below, so it can be measured.

  1. It is not clear the temperature of which point of the ingot section was calculated - surface / center? What's with the dip at the beginning of the graph?

Response:Thanks very much for the comment. The figure 4 (a) vertical coordinates have shown that the temperature of the ingot section is surface temperature. The dip at beginning of the graph is due to cooling of the foot roll section. The cooling intensity of the bloom is decreased after exiting the mold, resulting in the dip at the beginning of the graph.

  1. In the first paragraph of section 5.1. and in several other places in the text, conductivity values are given without units of measure.

Response:Thanks very much for the comment. The errors had been modified and were marked in red.

  1. As the main formula for calculating the electromagnetic force, the authors, with reference to [24], took the expression F≈B2f. This is not entirely clear. In [24] itself, the equation

T=0.25πσω B2R4L

But despite the fact that the main goal of the work under review indicated by authors as necessary to establish the effect of the difference in conductivity between the hardened crust and molten steel and the effect of the dimensions of the workpieces on the electromagnetic field, these parameters were not taken into account in the calculation of the force. Why?

Response:Thanks very much for the comment. Chapter 5.3 mainly considers the regulation of electromagnetic force changes at the center of the stirrer as the current frequency increases under the same billets size at 500A current intensity conditions. The reason for using the empirical formula F≈B2f is to compare the pattern of the effect of the change in current frequency on the electromagnetic force for the same billet sizes, so the equation T=0.25πσω B2R4L is not used.

  1. It can be seen from the text of the work that the calculations were carried out for a specific production line. However, there are no practical recommendations for production in the conclusion.

Response:Thanks very much for the comment. The conclusion had been modified and were marked in red.

Round 2

Reviewer 2 Report

Thanks to the authors for the answers. I think the work can be published.